# A Review of the Antiviral Activities of Glycyrrhizic Acid, Glycyrrhetinic Acid and Glycyrrhetinic Acid Monoglucuronide

**DOI:** 10.3390/ph16050641

**Published:** 2023-04-23

**Authors:** Jiawei Zuo, Tao Meng, Yuanyuan Wang, Wenjian Tang

**Affiliations:** 1Department of Pharmacy, The Second Affiliated Hospital of Anhui Medical University, Hefei 230011, China; 2School of Pharmacy, Anhui Medical University, Hefei 230032, China; 3Department of General Surgery, The Third Affiliated Hospital of Anhui Medical University, Hefei 230061, China

**Keywords:** glycyrrhizic acid, GAMG, glycyrrhetinic acid, antiviral, SARS-CoV-2

## Abstract

Licorice, a natural medicine derived from the roots and rhizomes of *Glycyrrhiza* species, possesses a wide range of therapeutic applications, including antiviral properties. Glycyrrhizic acid (GL) and glycyrrhetinic acid (GA) are the most important active ingredients in licorice. Glycyrrhetinic acid 3-*O*-mono-β-d-glucuronide (GAMG) is the active metabolite of GL. GL and its metabolites have a wide range of antiviral activities against viruses, such as, the hepatitis virus, herpes virus and severe acute respiratory syndrome coronavirus 2 (SARS-CoV-2) and so on. Although their antiviral activity has been widely reported, the specific mechanism of action involving multiple links such as the virus itself, cells, and immunity are not clearly established. In this review, we will give an update on the role of GL and its metabolites as antiviral agents, and detail relevant evidence on the potential use and mechanisms of actions. Analyzing antivirals, their signaling, and the impacts of tissue and autoimmune protection may provide promising new therapeutic strategies.

## 1. Introduction

Glycyrrhizic acid (GL), a tetracyclic triterpenoid saponin, has a broad spectrum of biological activities [1], especially antiviral effects [2]. As shown in Figure 1, GL is composed of two molecules of glucuronic acid and one molecule of glycyrrhetinic acid (GA). GA, the aglycone of GL, is one of the active metabolites of GL under the action of gut commensal bacteria. GA is also one of active ingredients in licorice, and possesses extensive pharmacological activities such as anti-inflammatory [3], antioxidative [4], and antiviral effects [5]. GL can be metabolized in the intestine or be transformed via enzymolysis to glycyrrhetinic acid 3-*O*-mono-β-d-glucuronide (GAMG). GAMG, a distal glucuronic acid hydrolysate of GL, has a higher bioavailability and stronger physiological functions than GL, including antitumor, antiviral and anti-inflammatory activities [6,7]. In particular, GL and its metabolites have antiviral infection properties and improve symptoms caused by viral infections.

Viruses usually include plant viruses, animal viruses and bacterial viruses. It is an organism with a special structure, usually containing only DNA or RNA, and parasitic in living cells, multiplying by self-replication. However, many viruses have a strong ability to invade, destroy, and even endanger the health of humans, animals, and plants. For example, the new SARS-CoV-2 virus that broke out in 2019 has been an ever-present threat to public health worldwide, and has already resulted in millions of dead. The constant mutation of the virus has brought great difficulties to epidemic prevention and people’s treatment. In addition, there are many common viruses in daily life, such as hepatitis viruses, influenza viruses, and herpesviruses.

Viral proliferation is a complex process, and the antiviral effect of natural products may influence the entry (attachment, penetration, intracellular trafficking, and uncoating), gene replication, and exit (assembly and maturation, and release) [8]. The antiviral mechanism of GL and its metabolites was not fully clear. Research reports indicated that GL can inhibit viral replication [5], regulate the fluidity of the plasma membrane, and affect the virus’s function of entering the cell and stabilizing the membrane [9,10]. For instance, acute or chronic hepatitis patients with elevated alanine aminotransferase were effectively treated with diammonium glycyrrhizinate enteric-coated capsules and diammonium glycyrrhizinate injections. GL can significantly reduce steatosis and necrosis of liver cells [11,12], inhibit liver fibrosis and inflammation, and promote cell regeneration. These also have been widely used as new inhibitors of the human immunodeficiency virus [13], Epstein-Barr virus (EBV) [14], SARS-related coronavirus [15] and Zika virus [16,17]. The bibliometric analysis showed that GL is most widely studied for antiviral in licorice [16,18]. GL may be a natural candidate for the treatment of the SARS-CoV-2 infection, as it has been reported to bind to multiple proteins of the virus, including the S protein, 3CLpro, angiotensin converting enzyme 2 (ACE2), etc. [19]. GL and GA greatly reduced inflammation through the interferon (IFN-γ), which meant that GL and GA have very important antiviral properties [20].

In this review, two investigators conducted an independent literature search using the PubMed, Embase, Cochrane Library, and Web of Science databases from the databases’ inception to 31 July 2022 (Jiawei Zuo and Tao Meng). The following key terms were used: “glycyrrhizic acid”, “glycyrrhetinic acid”, “Glycyrrhizin monosaccharides”, and “antiviral”. The cited references for the relevant systematic reviews and conference proceedings were manually cross-checked to identify additional literature. Articles pertinent to antiviral activity of glycyrrhizic acid and its metabolites were selected for this review. Searches were not restricted by language, country, or publication date. Articles were initially screened based on the title and abstract reading, and then full texts of potentially relevant publications were obtained and reviewed by two authors independently (Jiawei Zuo and Tao Meng) to determine the publications’ eligibility. Any discrepancies were resolved by discussion, and a third assessor arbitrated any disagreements.

The purpose of this article is to comprehensively review the safety and efficacy of GL and its metabolites for antiviral effects. Complex mechanisms of action are elaborated to highlight major therapeutic targets that may help reduce the damage caused by various viruses. This article provides a narrative and critical review of the clinical efficacy and pharmacological effects of GL and its metabolites in various viral infections, keeping clinicians and researchers informed of the latest results and main roles. In addition, it provides an overview of some complexes containing GL and its metabolites, and the application of novel delivery regimes that contribute to a better understanding of their use and optimization.

## 2. Anti-SARS-CoV-2 Activity

In 2019, the Coronaviruse disease (COVID-19) broke out, caused large-scale human infection worldwide, and brought serious health hazards and economic losses. So far, the virus has continued to mutate and spread, and it is still seriously endangering human health and safety. At present, the SARS-CoV-2 infection cannot be effectively inhibited after vaccination, and there is no effective drug for the treatment of the viral infection in clinical practice. It was previously reported that ACE2 was proven to be an important and special receptor for the SARS coronavirus [21]. Recent research showed that the receptor-binding domain of SARS-CoV-2 was specific for the human ACE-2 receptor, so the human ACE2 receptor acted as an important mediator of the SARS-CoV-2 infection [22,23].

SARS-CoV-2 is a β-coronavirus which contains three transmembrane proteins, spike protein (S), membrane protein (M), and envelope protein (E). SARS-CoV-2 also contains a large nucleoprotein (N) enveloped positive RNA genome [24]. GA has a good affinity for the spike protein (S) and master protease (Mpro) receptors of SARS-CoV-2 [25]. Since SARS-CoV-2 has an impact on multiple organs by attaching to the ACE2 receptor, patients may suffer obesity, dyslipidemia, diabetes mellitus, cardiovascular diseases, renal diseases, gastro-intestinal diseases and neurological diseases [26]. COVID-19 is the subject of intense research, with potential therapeutic drugs in various stages of testing, and with as many new therapeutic targets being investigated, such as the SSAA09E2 and CP-1 peptide that interfere with ACE2 recognition, and camostat and nafamostat that inhibit the type 2 transmembrane serine protease (TMPRSS2) [27]. Crucially, virion enters into target cells through the ACE2 receptor binding to a highly glycosylated spike protein trimer. Recently, GL was reported to have the ability to bind to ACE2, which can prevent SARS-CoV-2 infection.

Virtual screening also revealed that GL is effective against the target proteins of SARS-CoV-2 [28], serves a potential inhibitor of the ACE2-specific receptor binding domain (RBD) on the spike glycoprotein of the SARS-CoV-2 [29,30], or prevents SARS-CoV-2 from entering the host cells [31]. Further, GL can directly bind to the site of the SARS-CoV-2 spike interaction, interfere with spike ACE2 action, thus blocking binding and fusion events in the virus life cycle [32]. Additionally, GL interacts with the S protein and achieves the antiviral activity through mediating cell attachment and entry of SARS-CoV-2, blocking S-mediated cell binding [33]. The silicon docking research also proved the above views that GL and GA can directly interact with the ACE2, spike protein, TMPRSS2, and 3-chymotrypsin-like cysteine protease, key players in viral internalization and replication [34]. In vitro experiments have shown that GL is the most potent and non-toxic broad-spectrum anti-coronavirus molecule due to the disrupted interaction of the S-RBD and ACE2 by GL.

The various pharmacological activities of GL and its derivatives were summarized, and the structurally modified GL-related derivatives were further developed to make them less cytotoxic and more targeted [35]. Moreover, the interaction between GL and the envelope protein was revealed to be an effective inhibitor of the SARS-CoV-2 envelope protein at the molecular and structural levels [36]. In summarizing these studies, we can think that the inhibitory effects of GL on SARS-CoV-2 are through the following three aspects. Firstly, GL prevents virus replication and spread by interacting with ACE2, the potential receptor of SARS-CoV-2. Secondly, GL directly interacts with the spike protein on the SARS-CoV-2 envelope, thereby blocking binding and fusion events in the viral life cycle. Thirdly, GL acts as a potential envelope protein inhibitor of SARS-CoV-2, disrupting the function and structure of the virus.

The RBD in the SARS-CoV-2 nucleocapsid protein may be a potential drug target that plays a variety of key functions in the life cycle of the virus, especially viral replication. GL showed a higher binding affinity with target proteins, which may help block the important site within the RNA binding domain of SARS-CoV-2 viral nucleocapsid protein (N) and reduce the risk of infection in the host [37]. Viral RNA-dependent RNA polymerase (RdRp) remains the preferred target for COVID-19 prevention or treatment, and nucleoside analogues are the most promising RdRp inhibitors, with the limitation that nucleoside analogues can be removed by SARS-CoV-2 exonuclease (ExoN). GL has been identified as a potential RdRp inhibitor with ExoN activity, contributing to better antiviral effects [38].

In addition, virtual screening showed that GL had the best affinity towards key proteases of SARS-CoV-2, which played a pivotal role in mediating viral replication and transcription [31,39]. It had been reported that GL and GA potentially inhibited SARS-CoV-2 infection, while revealing that the target of GL was the nsp7 protein, and GA binds to the spike protein of SARS-CoV-2 [40]. There are various reports on the antiviral mechanism of GL and its derivatives, but most studies focus on ACE2. Murch pointed out that GL and its metabolites directly inhibited of expression of TMPRSS2, which is related to virus entry, then reduced the expression of ACE2 [18].

From a pharmacological point of view, GL showed therapeutic potential for COVID-19 through binding to ACE2, down-regulating proinflammatory cytokines, inhibiting intracellular reactive oxygen species (ROS) accumulation, suppressing high respiratory tract output, and inducing endogenous interferons [41]. The anti-inflammatory activity of GL may play a crucial role in the excessive inflammatory response after SARS-CoV-2 infection. Network analysis and protein-protein interactions showed that the GL’s key targets for COVID-19 may include the intercellular cell adhesion molecule-1 (ICAM1), matrix metalloproteinase-9 (MMP9), toll-like receptor 2 (TLR2), and suppressor of cytokine signaling 3 (SOCS3), and the inflammatory cytokine signals, growth factor receptor signaling, and complement system were the crucial pathways by pathway enrichment analysis [42]. In practice, the interesting clinical investigations in China have verified the effects of diammonium glycyrrhizinate-vitamin C tablets on the common COVID-19 pneumonia, and its metabolite GA, which is structurally similar to corticosteroids, may act as a glucocorticoid-like drug, helping to enhance immune regulation against cytokine storms and reduce inflammation [43]. GL does not strongly down-regulate proinflammatory cytokine activity as glucocorticoids but has an immunosuppressive effect [44]. The new combination of GL, vitamin C, and curcumin has the potential to modulate the immune response after CoV infection and inhibit excessive inflammation to prevent the onset of cytokine storms [45]. Due to benign safety and the hepatoprotective effect, GL is expected to be an antidote for numerous therapeutic drugs, improving liver damage after SARS-CoV-2 infection [46].

In more recent years, there have been an increasing number of studies on the activity of GAMG. During the process of the LPS-induced RAW264.7 cell inflammatory response, GAMG showed a higher anti-inflammatory activity than GL, which may be due to the stronger inhibitory effect of GAMG on IL-6, iNOS, and COX-2. In-depth research on the anti-inflammatory mechanism of GAMG found that it could block nuclear factor-κB (NF-κB) and mitogen-activated protein kinase (MAPKs) signaling pathways [3,47]. High intake of GL may disturb the ion metabolic balance and may cause multiple adverse effects in animals and humans [48]. Specially, GAMG contains a glucuronic acid in its structure, which has good polarity, enabling it to pass through hydrophobic and hydrophilic cell membranes, and then play a better active role [49].

The long-term medical GL and its preparations made GL a good candidate against SARS-CoV-2 [50]. The combination of GL and boswellic acid could significantly shorten the recovery time and reduce the mortality rate of hospitalized patients with moderate COVID-19 infection [51]. Increased ACE2 expression during pregnancy may also increase the susceptibility of pregnant women for the virus, and GL treatment as a safer alternative to antiviral may be a good strategy [52]. GL nanoparticles (GANP) targeting severely inflamed areas significantly improved the anti-coronavirus therapeutic effect of GL through increased GANP biocompatibility, increased accumulation in the lungs and liver, enhanced the permeability and retention (EPR) action in the assimilated SARS-CoV-2-infected mouse model [53]. The National Health Commission recommended licorice and its active components for treating of COVID-19 infectious pneumonia [41]. Moreover, some derivatives of the GL have been manifested so that they have multifold antiviral activity [54].

The antiviral activity of GA has also been extensively studied. Due to high safety as a natural sweetener, GA is widely used as a clinical therapeutic [55]. At present, due to the lack of effective vaccines and therapeutic drugs, GL and its metabolites have the therapeutic potential for developing products as antiviral agents. Summarizing the above studies, GL and its metabolites are widely used and generally safe compounds that we consider investigating for primary prevention. It does not reduce the risk of infection, but it may mitigate the severity of the disease and reduce the burden of the medical care process. We refer to the life cycle of SARS-CoV-2 and the possible inhibitory targets of antiviral drugs mentioned in Frediansyah et al., and we summarize the mechanism by which GL and its metabolites may affect SARS-CoV-2 as shown in Figure 2 [56].

## 3. Hepatitis Virus

The liver is an organ with mainly metabolic functions in the body and plays a role in deoxidation, storage of glycogen, and synthesis of secreted proteins. Liver disease accounts for approximately 2 million deaths per year worldwide, due to complications from cirrhosis, and viral hepatitis and hepatocellular carcinoma (HCC). GL and its derivatives may have a therapeutic value in the hepatitis disease caused by viral infections. Some glycyrrhizin treatment mechanisms and references for various hepatitis viruses are shown in Table 1.

### 3.1. Anti-Hepatitis B Virus

The hepatitis B virus (HBV) infection is one of the major public health problems worldwide, about 30% of the world’s population shows serological evidence of past or present infection [78]. It is generally believed that the degree of liver damage and virus control depend on the complex interactions between viral replication and the host’s immune response. GL has also been used for years to treat chronic hepatitis B and improve liver function for many years. It had no toxic effect on host cells, but did moderately inhibit HBV production [79]. GL was highly maintained in hepatocytes and could suppress intracellular secretion and transport of HBsAg in PLC/PRF/5 cells [60]. The rationale may involve altering the expression of HBV-related antigens and inhibiting the sialylation of HBsAg. The inhibition of HBsAg by GL was also shown in cells treated with GL by the reduced accumulation of labeled HBsAg [59]. The sialylation of HBsAg was also inhibited by GL and accumulated in the cytoplasmic vacuoles near the Golgi apparatus. These findings hint that GL can increase HBV-related antigen expression and immune recognition by interfering with HBsAg transport and modification, thereby improving liver function in patients with hepatitis B. In fact, GL not only inhibits HBV antigen but also may promote the recovery of patients with hepatitis B through anti-inflammatory activity, thymic T cell activation [62], and immunological regulation [63].

In 2017, researchers were surprised to find that GA had a synergistic effect with entecavir for the therapy of HBV infection [66]. GA influenced entecavir at the cellular and subcellular levels of the liver by inhibiting the multidrug resistance protein 4 and the efflux transporters breast cancer resistance protein. In fact, GA had not affected the plasma pharmacokinetics of entecavir but increased its distribution in the nucleus and cytoplasm of hepatocytes, thereby enhancing antiviral efficacy. Intravenous GL rapidly improved serum transaminases in patients with acute exacerbation of chronic hepatitis B [80]. One study reported that a cancer patient, with an HBV infection that may have been caused by a blood transfusion, was administered entecavir and GL for antiviral treatment. Eventually, her liver tests were close to normal limits [64]. Similarly, a chronic HBV carrier who developed HBV hepatitis after receiving conventional-dose chemotherapy for non-Hodgkin’s lymphoma had suppressed HBV-DNA levels and normalized transaminase levels after receiving lamivudine in combination with GL [65]. Those patients were infected with HBV again in a special period with other diseases; their immune functions changed and resistance to the disease declined. It is unclear how the combination of different drugs works; however, nucleoside antiviral drugs combined with GL and its metabolites in the treatment of HBV may play a vital role in enhancing the antiviral effects of the drugs or maintaining liver function.

### 3.2. Anti-Hepatitis C Virus

The hepatitis C virus (HCV) infection is an important cause of chronic hepatitis C, cirrhosis and HCC worldwide [81]. GL inhibited HCV full-length viral particles and exerted anti-inflammation and immunity regulation [82]. Currently, there is no protective vaccine or effective eradication therapy. Biochemically, the role of GL in its action may involve stabilizing cell membranes, while GL treatment can alleviate liver necrosis and inflammation to a certain extent. Interestingly, in HCV-infected Huh7 cells treated with GL, the extracellular infectivity was decreased, but the intracellular infectivity was increased, which may be because GL inhibits PLA2G1B and affects HCV release [12]. Furthermore, GL possibly acted as a radical scavenger, decreased cell membrane permeability, prevented membrane penetration of viral particles, and reduced damage to liver tissue [75]. Research results proved that clinically appropriate doses of stronger neo-minophagen (SNMC) protected mitochondria from oxidative stress generated by HCV proteins and iron overload, and preventing hepatic steatosis [11]. GL can synergize with interferon and inhibit HCV core gene expression and HCV full-length viral particles [74].

Furthermore, GL intravenous injection obviously decreased the incidence of HCC in HCV-infected patients [72]. GL along with interferon may be more effective in treating HCV infection. SNMC is commonly used in patients who are resistant to IFN or who have side effects with IFN therapy. It is noteworthy that, although the mechanism of action of GL on the liver is unknown, normalization of ALT levels in HCV infection may indicate reduced liver tissue damage. Clinical reports indicated sixty percent of hepatitis C patients who take a six times weekly administration of GL had an obvious decrease of ALT several weeks later [67]. Researchers synthesized a series of GA metabolites and confirmed that they all exhibited better anti-HCV activity than GA by MS and NMR spectroscopy [76]. GL and its metabolites combined with antiviral drugs and new treatments can effectively inhibit HCV infection and reduce the incidence of liver carcinogenesis.

### 3.3. Anti-Hepatitis E Virus

The hepatitis E virus (HEV) is an enterically transmitted virus [83]. Globally, HEV infection is usually an acute self-limiting disease [84] and the most common reason for acute viral hepatitis. The effect of glycyrrhizic acid and its metabolites on the HEV activity has been poorly reported, with one early study reporting a reduction in the total normalized bilirubin, ALT and AST levels [77]. It may suggest GL shortened the duration of illness and the fatal complications of acute liver failure due to HEV infection.

## 4. Herpesvirus Activity

Herpesviruses are large, diverse and have a double-stranded DNA (dsDNA) molecule wrapped in a rigid protein shell. They contain a wide variety, such as herpes simplex virus 1 (HSV-1) and HSV-2, EBV, varicella zoster virus (VZV), human cytomegalovirus (HCMV), human herpesvirus-6 (HHV-6), Human herpesvirus-7 (HHV-7), and Kaposi’s sarcoma-associated herpesvirus (KSHV) [85]. In general, the herpesviruses are ubiquitous in humans, and can establish long-term latent infections or can cause various diseases from infection. In recent years, herpesvirus infections have become the main cause of death for people with impaired immune function. Glycyrrhizin, a component of the licorice root extract, is also active against various herpesviruses [5].

### 4.1. Anti-Herpes Simplex Virus

HSV-1 can cause severe diseases among people with low immunity. Under normal circumstances, T cell activity that is related to cell-mediated immunity plays an important defense against herpesvirus infections. Researches indicated that the mortality rate of special pathology mice exposed to HSV decreased from 80% to 5% after GL treatment, which significantly improved survival in HCV-infected mice [86]. These results suggested that GL may increase the resistance of individuals with low immunity to opportunistic herpesvirus infections by inducing CD4+ anti-suppressor T cells. In some cases, viruses could inhibit autophagy activation, thereby creating favorable conditions for their own replication and pathogenicity [87]. GL and rapamycin also induced the production of high amounts of Beclin1, which can be used as a strong activator of autophagy and further establish resistance to HSV1 replication [88]. GL and lysozyme or lactoferrin both showed synergistic anti-virus effect on Vero cells infected with HSV-1 [89]. The inhibition of HSV-1 by GL in vitro assays demonstrated an obvious synergistic effect. Some studies suggested that the combined use of more than one drug may increase antiviral activity, reduce viral resistance and decrease adverse reactions [89,90].

GL and its derivatives were capable of hindering the growth and cytopathic effects of HSV [5,91]. In addition to the above, GL and its modified compounds have antiviral activity against HSV and can permanently inactivate the virus [5,92]. Animal studies demonstrated that GL and its derivatives also reduced the viral activity and mortality of HSV encephalitis [93]. Increased cell adhesion is an important pathological mechanism leading to an inflammatory response during HCV infection. GL significantly reduced the stress and adhesion between rat cerebral capillary vessel endothelial cells (CCECs) and polymorphonuclear leukocytes (PMNs) during HSV infection, thereby reducing the inflammatory response of HSV [94]. In particular, the anti-HSV-1 activity of the water extract of licorice roots was higher than that of the alkaline extract, and mechanisms of action of water and alkaline extracts against HSV-1 may be different [95]. GL treatment of mice with herpetic encephalitis caused by inoculation of HSV-1 onto the cornea increased their survival by an average of 2.5-fold, while reducing HSV-1 replication in the brain by nearly half of the control [96]. Furthermore, GL showed better inhibition of the HSV-1 virus replication in vivo than in vitro, and the effect of GA was 10 times higher than that of GL, so in vivo anti-HSV-1 activity of oral GL can be reasonably attributable to GA produced by intestinal bacterial hydrolysis [97]. Besides, 18α-GA acted as a gap junction communication inhibitor, reducing ganciclovir toxicity in bystander cells transfected with HSV-1 [98].

### 4.2. Anti-Varicella Zoster Virus

VZV is a globally infectious human virus that causes chickenpox and is usually spread through broken skin or respiratory droplets [99]. Under certain conditions, GL inactivated more than 99% of VZV particles within a certain concentration range [100]. Perhaps GL inhibited viral replication in the very early stages of the replication cycle, i.e., infiltration or uncoating of the virions. Furthermore, when some anti-herpes drugs, which included acyclovir and adenine arabinoside or human natural IFN-β, were used in combination with GL, it had a slightly synergistic or an additive effect on VZV replication. However, compared with acyclovir and IFN-α2a, crude GL had a low antiviral activity against VZV [101].

### 4.3. Anti-Epstein-Barr Virus

EBV is a well-known pathogen of infectious mononucleosis, and many diseases are often related to EBV in etiology. Many types of cancer are accompanied by dysregulated sumoylation processes [102,103,104]. EBV latent membrane protein 1 (LMP1) regulates the cell sumoylation process and contributes to its carcinogenic potential in EBV-related malignancies. GL could inhibit the sumoylation processes (without affecting the ubiquitination processes) in lymphoblastoid cell lines transformed with EBV expressing latent membrane protein 1, limiting cell growth, and increase apoptosis in various cell lines [105]. So, investigators proposed that GL did reduce the ability to infect other cells to a certain extent, and it might serve as a potential drug for treatment of EBV-associated lymphoid malignancies. An early report suggested that GA showed the remarkable inhibition of an early antigen of EBV (EBV-EA) [106]. The antiviral effect of GL was not directly involved with inactivating virus particles, nor inhibiting virus adsorption, but inhibiting EBV-DNA replication (seeming to be at the penetration phase) and viral antigen synthesis [107]. However, GL had no effect on the EBV-DNA of various cells that either spontaneously produced virus or latently infected cells that did not produce virus. Therefore, the mode of action of GL is different from that of the nucleoside analog target, which is a virus-encoded DNA polymerase. The isomers of GL (18α-GL and 18β-GL) were also effective, but its sodium salt lost antiviral activity. GA was more active against EBV, but the cytotoxicity also increased [14]. The amino acid residue contained in the carbohydrate portion of GL was active against EBV, such as Ala-OMe glycopeptide, Leu glycopeptide, and Glu (OH)-OMe glikoproteiny. In particular, when the hydroxyl group on the molecule is replaced by a methoxy group, the polarity of the molecule changes and the antiviral activity disappears completely. On the basis of the GL core structure, different amino acids may be connected to produce different antiviral activities.

### 4.4. Anti-Kaposi’s Sarcoma-Associated Herpes Virus

KSHV is the cause of Kaposi’s sarcoma, multicentric Castleman disease, and primary exudative lymphoma. Patients treated with GL had lower KSHV DNA levels than those without therapy [108]. GL, which treated cells latently infected with KSHV, reduced viral protein synthesis during incubation and induced apoptosis in infected cells [109]. It induced cell death in KSHV-infected cells. The specific pathways were down-regulated from latency-associated nuclear antigen (LANA) expression and up-regulated from viral cyclin expression [110].

The activation of any TLR would result in the production of interferon and the upregulation of cytokines and chemokines [111]. However, KSHV vIRF1 also inhibited IFN-induced gene functions, such as the TLR4 transcription and downregulation [112]. Recently, the cytoplasmic variant of LANA can directly bind to cGAS to inhibit the cGAS-STING DNA sensing pathway [113]. Members of the cytoplasmic DNA sensing pathway included cyclic GMP-AMP synthase and STING, and the cGAS-STING pathway seemed to be able to detect KSHV in the initial stage of infection and latency reactivation of multiple cell types [113,114]. Importantly, GL downregulates the expression of LANA-1. In this way, GL can indirectly attenuate the outer protein ORF45, block IRF7 phosphorylation and the activation of the type I IFN, and reduce the inhibitory effect of KSHV-encoded viral interferon regulatory factor (vIRF) on the activation of downstream IFN and NF-κB. GL downregulation of LANA-1 was related to changes in mitochondrial membrane potential, which transferred apoptosis-inducing factors (AIF) to the nucleus, causing DNA fragmentation and apoptosis. In addition, GL led to a higher phosphorylated (active) p53 content and cell cycle arrested at the G1 checkpoint. GL has the ability to promote the expression of cell cycle protein (v-cyclin), but has no effect on viral FLICE inhibitory protein (vFLIP). In short, GL reduced LANA levels, inhibited inflammation, induced interferon, activated p53, and increased ROS and mitochondrial dysfunction, resulting in KSHV-infected cells stopping proliferation, apoptosis, etc. Through this series of actions, GL can block virus reproduction and enhance immune resistance. Part of the mechanism of action of GL on KSHV is shown in Figure 3. Interfering with the replication of KSHV genes during the latent period and reducing the occurrence of induced tumors may have therapeutic significance for eliminating potential KSHV infection.

GL might affect the maturation of mRNA encoding LANA and selectively inhibit the growth of KSHV-infected lymphocytes [115]. The study had shown that GL destroyed the RNA polymerase II (RNAPII) complex of CTCF-cohesin binding sites that accumulated in the first intron of the latency transcript. This resulted in an altered enrichment of the RNAPII pause complex, and the pause SPT5 and NELF-A at the cohesive binding site of CTCF. With new discoveries, GL more commonly causes the loss of cohesion in the sister chromatid of the cellular chromosomes. GL interacted with the cellular proteins SMC3 and SPT5, but not with their relationship partner RAD21 and RNAII. This loss caused by GL will lead to lessen the mRNA production and the defect of sister chromatid condensation, which is important for the stability of virus and cellular chromosomes.

## 5. Anti-Human Immunodeficiency Virus Activity

Clinical practice has shown that GL treatment of patients with hemophilia or with acquired immunodeficiency syndrome could increase the level of the p24 antigen of the human immunodeficiency virus-1 (HIV-1), suggesting that GL may inhibit HIV-1 replication in vivo [116]. GL not only inhibits HIV but also induces interferon production, enhances natural killer C effects, and increases the number of CD4-positive T lymphocytes to effectively prevent the development of acquired immune deficiency syndrome (AIDS) [117]. Furthermore, it might also inhibit HIV replication by reducing the activity of protein kinase C in Molt-4 cells or interfering with virus adsorption into the cells [118]. GL completely inhibited the expression of an HIV-specific antigen and the cytopathic effect of HIV in MT-4 cells. Protein kinase C phosphorylates the CD4 molecule, which could be a key link in HIV-1′s interaction with the CD4 receptor, and a protein kinase C inhibitor, 1-(5-isoquinoline-sulfonamide)-2-methylpyrrole xylene salt, inhibits the replication of HIV-1 in lymphocytes. In fact, electron microscopy showed that protein kinase C inhibitors did not destroy the combination of HIV and CD4+ cells but caused a significant accumulation of virus particles on the cell surface while inhibiting the infectivity of the viruses. Therefore, GL may play a role in protein kinase C inhibition, interfering with HIV binding to receptors and impeding the normal HIV replication process.

There are some clinical data showing that in cultures of 12 patient peripheral blood mononuclear (PBM) cell samples infected with a slow-replicating nonsyncytium-inducing variant of HIV (NSI-HIV), GL inhibited HIV replication by more than 90% [119]. Differences in chemokine receptors are vital to HIV cell tropism, such as CCR5 that is the main coreceptor for HIV (R5 HIV) macrophage strains [120]. The process by which NSI-HIV enters macrophages is regulated by β-chemokines. The mechanism by which GL inhibits NSI-HIV replication may be related to the stimulation of β-chemokine production to compete with chemokine receptor-mediated HIV-infected cells. After treatment of human PBM cells with 1-methyladenosine (MA), R5 HIV replication occurred, and soluble factors CCL2 and IL-10 were generated [121]. GL inhibited the production of IL-10 and CCL2 in PBM/MA, thereby reducing the expression levels of CCR5 mRNA and inhibiting the replication of R5 HIV. PMNs exposed to R5HIV produce IL-10 and CCL2, and this process enhanced R5 HIV replication in macrophage cultures [122]. When a certain concentration of GL was added to the cocultures, the inhibition of R5 HIV production macrophages cocultured with PMNs exceeding 95%. GL inhibits the expression of CCR5 by reducing the production of IL-10 as well as CCL-2 and reduces the replication of R5 HIV greatly in the end.

After the virus particles attach and adsorb to the specific receptor on the host cell, the enveloped virus must pass its own lipid bilayer envelope coating and the plasma cell membrane lipid bilayer fusion layer through the fusion pore to enter the host. The fluidity of the HIV-1 envelope was always lower than that of the plasma membrane because the cholesterol/protein ratio of the viral envelope was higher than that of the plasma membrane [9,123]. Decreased membrane fluidity inhibited the formation of viral synapses associated with the inhibition of intercellular fusion. GL inhibited the infectivity of HIV particles by reducing the fluidity of the membrane [9]. Interestingly, GL also inhibited UV-induced HIV gene expression [124]. In particular, alkaline and water extracts of licorice have opposite activities against HIV or HSV [95]. There are definite differences in the ingredients and contents of licorice root water or alkaline extract, and the mechanism of action against HSV and HIV is also dissimilar. The triterpene or peptide part alone has weak activity on HIV-1 Env-mediated cell fusion, but through the click chemistry method, the biologically active triterpene saponin is connected to the helix region-containing binding domain on the peptide, generating strong synergy [125]. Structural modifications of GL inhibited the HIV-1 virus-specific protein p24 more than itself [126]. Some compounds with a structural modification that includes the GL nucleus may provide more therapeutic strategies for the treatment of HIV or other viruses.

## 6. Anti-Influenza Virus Activity

The influenza virus and respiratory syncytial virus infection have become the main killers for viral respiratory diseases, and Chinese herbal medicines can inhibit virus reproduction, enhance human immunity and so on, which have good antiviral activity. The earlier studies showed that the treatment of mice infected with 10–50% of the lethal dose of the influenza A2 virus with GL exhibited promising therapeutic efficacy [127]. GL can directly reduce the mortality of influenza virus-infected animals. The actual research found that the influenza A2 virus-infected mice treated with GL had significantly an improved internal pathology of lung tissue, meanwhile the virus titers in the experimental animals were also at a low level. It indicated that GL had no effect on the replication and survival of influenza A2 in vitro, but it significantly improved animal pathological and histological damage. Similar findings showed that GL treatment significantly reduced the number of human lung cells and CCID50 titers infected with influenza virus A [128]. Additional evidence suggested that the effect of GL on influenza virus was limited to the early stages of viral replication and ruled out that GL directly inhibited influenza A virus particles or interfered with virus-receptor binding interaction. The GL antiviral activity was mediated through the interaction with cell membranes, reducing endocytosis activity and virus uptake. These can further improve cytopathy effect, reduce viral RNA content in intracellular as well as cell supernatants, and reduce viral hemagglutination titers [128].

GL had little effect on H5N1 replication and apoptosis of infected cells to a certain extent but diminished the migration of monocytes to the supernatant of A549 cells infected with H5N1 [129]. GL, at an effective concentration, substantially inhibited the expression of the H5N1-induced proinflammatory molecules such as CXCL10, CCL2, CCL5, and IL-6. The influenza A virus infection activated ROS, while antioxidant inhibitors were able to inhibit virus-induced proinflammatory gene expression and influenza A virus replication [130]. Could GL act as an antioxidant inhibitor in virus-infected bodies? GL ultimately interfered with the replication of the highly pathogenic H5N1 influenza A virus and the expression of H5N1-induced proinflammatory genes by reducing ROS formation and the expression of NF-κB, JNK, and P38. However, the effectiveness of the mechanism of GL on influenza virus is often more than MAPK and NF-κB. Besides, the presence of a functional high mobility group protein B1 (HMGB1) DNA binding site is necessary to enhance the replication ability of the influenza virus. Viral nucleoprotein in the nucleus of infected cells was associated with HMGB1, promoted viral growth, and enhanced viral polymerase [131]. GL attenuated the binding ability of HMGB1 to DNA and affected the activity of the influenza virus polymerase. Coincidentally, GL derivatives may also have anti-influenza virus activity. For instance, the introduction of cysteine or phenylalanine in the carbohydrate part of GA could be the most efficient to improve the activity of anti-A/H1N1/pdm09 influenza virus [132]. To date, the introduction of various substituents into GL and GA molecules, along with novel materials such as nanoparticles or micelles that transport compounds, has increased the ability to inhibit the reproduction of various influenza A and B strains [133].

The triterpenoid saponin isolated from licorice contained a saccharide moiety and glucuronic acid or galacturonic acid and showed good inhibitory activity against the H1N1 influenza virus in MDCK cells [134]. Moreover, triterpene (GA and its derivatives)-sialic acid conjugates showed distinct inhibition of influenza virus proliferation and influenza virus sialidase subtypes [135]. It was subsequently found that GA combined with ribavirin showed a good synergistic effect on the survival of viral pneumonia in mice established by inoculation with the H1N1 influenza virus [136]. This combination significantly inhibited lung consolidation, resulting in a conspicuous reduction in viral titers in virus-infected lungs, and potent inhibition of the production of related pro-inflammatory cytokines. The combined use of GL and antiviral drugs, and the structural modification of GL both provide new strategies for improving antiviral activity. After treatment of MDCK cells infected with the influenza virus with Ma-Xing-Shi-Gan-Tang containing GL, viral RNA and protein synthesis were affected [137]. It destroyed the surface structure of the virus, prevented the virus from entering the stage, and involved the regulation of the PI3K/AKT signaling pathway. The complex containing GA also provides a new idea for antiviral treatment. Similarly, influenza viruses not only affect human life and health, but also harm animal health. The researchers used a hem-agglutination inhibition test to conclude that glycyrrhizinate caused fewer positive reactions in animals treated with H1N1 and H3N2, showing a lower rate of viral transmission. To a certain extent, GL stimulated immune responses against pig influenza, as measured by viral shedding [138].

## 7. Other Anti-Viral Activities

Most reports indicate that antiviral activity is one of the broad pharmacological activities of GL. GL is only active at high non-cytotoxic concentrations, but different concentrations of GL can effectively inhibit the replication of flaviviruses [139]. Likewise, GL decreased the production of Semliki Forest virus and Chikungunya virus, effectively reducing viral replication [140]. The application of GL to the model of coxsackievirus B3-induced myocarditis did not change significantly the virus titer [141]. However, the expression of myocardial pro-inflammatory cytokines, such as TNF-α, IL-1β, and IL-6, were effectively reduced at both mRNA and protein levels. The reason might be that it effectively inhibited coxsackievirus B3-induced NF-κB activity by intercepting the degradation of inhibitor IkBk. Collectively, GL has a profound effect on myocarditis caused by coxsackievirus B3, and its weight loss status is improved, while increasing the remission of serum enzyme levels, reducing myocardial inflammation, and improving survival rates. Differently, GL directly inactivated coxsackievirus A16, and its anti-enterovirus 71 effect was related to viral cell entry events, all blocking virus replication [142]. The activity of GL at different concentrations against coxsackievirus A16 and enterovirus 71 was significantly different, and the same concentration of it reduced the production of infectious coxsackievirus A16 and enterovirus 71 to varying degrees.

GL neither directly inactivated porcine reproductive and respiratory syndrome virus (PRRSV) particles nor inhibited the release of PRRSV. Among them, both PRRSV proliferation and the expression of PRRSV-encoded proteins were reduced. In terms of the mechanism, GL mainly inhibited the infiltration phase and had little influence on the PRRSV adsorption or release steps [143]. GL also moderately reduced porcine epidemic diarrhea virus (PEDV) infection in Vero cells by affecting viral entry and replication, but not viral assembly and release. It could be used as an immunomodulator against PEDV infection, reducing the level of proinflammatory cytokine mRNA [144]. In-depth research on the molecular direction showed that GL inhibited virus infection and involved the HMGB1/TLR4-MAPK pathway. During infection, TLR4 itself was unchanged, but TLR4 played an important role in regulating PEDV infection and proinflammatory cytokine expression. GL inhibited p38 phosphorylation by modulating the TLR4 receptor but not Erk1/2 or JNK phosphorylation. Surprisingly, mutation or knockdown of HMGB1 also reduced p38 phosphorylation. In summary, it inhibited PEDV infection and reduced the secretion of proinflammatory cytokines through the HMGB1/TLR4 MAPK p38 pathway [145]. When porcine parvovirus infected swine testis cells, diammonium glycyrrhizinate (DG) had a direct anti-porcine parvovirus effect in vitro, while the viral infectivity decreased [146]. DG potently inhibited the replication of porcine deltacoronavirus (PDCoV) strain SD2018/10 in LLC-PK1 cells. DG not only affected viral attachment but also reduced viral RNA in cell lysates early in PDCoV replication. LiCl and DG increased PDCoV-induced apoptosis. However, when PDCoV-infected cells were treated together, there was no significant difference in viral mRNA levels and no synergy between the two [147]. In addition, additional time experiments and antiviral tests demonstrated that dipotassium GL inhibited viral replication and promoted the transformation of Marek’s disease virus-infected cells instead of inactivating its particles in a nontime alternative way [148].

GL can effectively inhibit the cytopathic effect (CPE) induced by dengue virus 2 and reduce the infectivity of the virus at the same time [149]. The introduction of amino acids or their methyl esters and aromatic hydrazide residues would have a huge impact on antiviral activity. Modifications are made by introducing long-chain amino acids into the GA structure to keep the triterpene COOH group free. Continued discovery of amino acid ester conjugates of GL are screened for extended studies against ZIKV infection [16].

Carbon dots (CDs) are increasingly used in the field of antiviral research. This was a new attempt to combine CDs with GL to develop not only high biocompatibility but also excellent antiviral activity. Using PRRSV as a model, GL-based carbon dots (GL-CDs) activity detection was carried out [150]. Detailed investigations revealed that GL-CDs inhibit PRRSV proliferation through targeted invasion and replication. Additionally, GL-CDs regulated the level of interferon-stimulated genes to suppress viral infection, inhibited the accumulation of ROS in cells caused by PRRSV infection, and regulated the expression of certain host limiting factors, such as DDX53 and NOS3, those related to PRRSV proliferation. Researchers used rotavirus as a model and developed a medium-flux screening analysis method, which detected that 18β-GA activated NF-κB and induced IL-8 secretion, exerting antiviral effects [151]. The 18β-GA did not directly inactivate rotavirus particles. After virus adsorption, the addition of 18β-GA treatment to the infected culture mostly reduced rotavirus production. The 18β-GA interfered with one or more links after virus entry, and ultimately affected virus replication [152].

## 8. Prospective View

GL can treat the SARS-CoV-2 infection through interacting with ACE2 to prevent the viral replication and spread, interacting with the spike protein to block the viral binding and fusion, inhibiting the envelope protein to disrupt the viral function and structure, and targeting the RNA binding domain of nucleocapsid protein to prevent the viral replication. GL can also exert the therapeutic potential for COVID-19 through down-regulating pro-inflammatory cytokines, inhibiting ROS accumulation, decreasing thrombin, suppressing high respiratory tract output and inducing endogenous interferon. Physiology-based pharmacokinetic studies indicated that GAMG can be absorbed quickly and effectively, and exert better pharmacological activities than GL. GAMG exhibited benign drug-like properties due to higher blood concentration and higher tissue distribution in the lungs and kidney [153]. GAMG can inhibit COVID-19 virus and remedy COVID-19 associated pneumonia.

Further, molecular docking showed that GAMG binds to the non-structural protein 3CLpro of SARS-CoV-2, but GL and GA do not, indicating that GAMG may have a special and good activity for inhibiting SARS-CoV-2 (Figure 4A). Moreover, comparing the superposition diagram of GA and GAMG in the pocket, GA and GAMG fit well together as a whole, and that there is no major conformational change in the molecule as a whole triggered by the monosaccharide molecule partially preferring the hydrophilic pocket (indicated by the red dashed line). In contrast, in terms of the hydrophilic pocket occupied by the monosaccharide molecule (indicated by the red dashed line), the disaccharide molecule of GL may not fit in the hydrophilic pocket due to the space size factor, thus affecting activity (Figure 4B). It can be seen from Figure 1 that the GL structure has two molecules of glucuronide acid, and the molecular polarity is very strong and cannot enter the hydrophobic cell membrane, while the GA structure does not contain glucuronic acid, and the molecular polarity is too weak to enter the hydrophilic cell membrane. GAMG containing a single glucuronic acid differs from the other two in that the distal cleavage of glucuronic acid allows the molecule to have a good and suitable polarity and can smoothly pass through both hydrophilic and hydrophobic cell membranes. Therefore, the enhanced permeability of GAMG on the cell membrane may be beneficial to improve the absorption of molecules and binding to proteins, thereby improving bioavailability and biological activity.

The understanding of the interactions between GAMG, GL, GA and infectious factors can improve the remedy of COVID-19 patients and reduce the risk of antiviral resistance [27]. All in all, we speculate that GAMG may have better efficacy than GL and GA in the therapy and/or prevention of SARS-CoV-2 infection.

## 9. Conclusions

GL and GL-containing drugs have been widely used in clinical practice. GA and GAMG are the active metabolites of GL. GL and GA exhibit antiviral activity by affecting the proliferation and pathogenic effects of various viruses. Here, we summarized the action mechanism of GL, GA and GAMG against various viruses, including SARS-CoVs, hepatitis virus, and herpesvirus. GL and GA not only apply direct chemotherapy to inhibit various pathogenic viruses but also can interact with cell membranes or exert joint immunity with the host. GL can suppress gene expression of potential oncogenic viruses to treat severe and incurable chronic viral diseases, such as Kaposi’s sarcoma. GL can treat SARS-CoV-2 during and after virus adsorption through inhibiting virus adsorption and penetration. GL exerted the therapeutic potential for SARS-CoV-2 by combination with ACE2, down-regulating pro-inflammatory cytokines, decreasing the accumulation of ROS, inhibiting thrombin and airway exudates, and inducing endogenous interferon. As promising antiviral drug candidates, GL, GA and GAMG can inhibit virus replication and eliminate latent virus to combat the increasing variety and destructive power of viruses. Interestingly, pharmacokinetics and molecular docking showed that GAMG with benign drug-like properties can more efficiently target ACE2 and remedy SARS-CoV-2 infection than GL. Thus, GAMG and GL are expected to become broad-spectrum antiviral agents and be widely used in clinical treatments.

## Figures and Tables

**Figure 1 pharmaceuticals-16-00641-f001:**
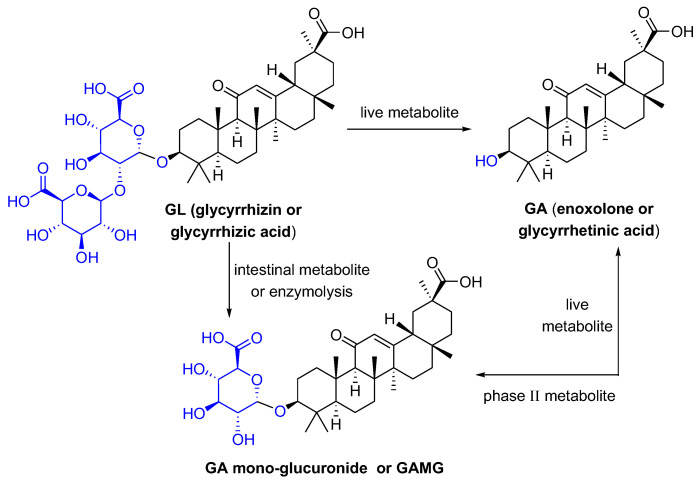
The structures of GL and its metabolites GA and GAMG.

**Figure 2 pharmaceuticals-16-00641-f002:**
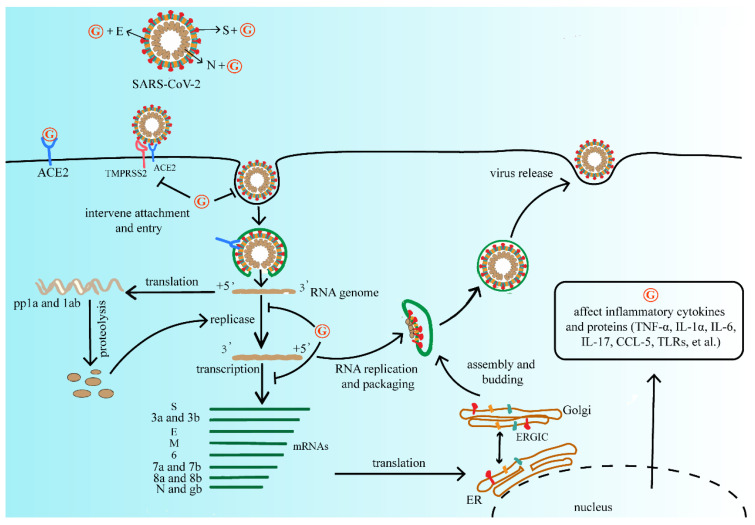
The mechanism of GL and its metabolites on the replication process of SARS-CoV-2. They may affect viral protein function, or compete with the ACE2 receptor of SARS-CoV-2, further interfere with virus adsorption, prevent virus penetration into the cell, and inhibit virus biosynthesis, at the same time decrease virus release, and finally reduce the process of viral infection. Abbreviations: G, GL and its metabolites; ACE2, angiotensin converting enzyme 2; TMPRSS2, type 2 transmembrane serine protease; SARS-CoV-2, severe acute respiratory syndrome coronavirus 2; pp1a, serine/threonine protein phosphatase 1; TNF-α, tumor necrosis factor-α; IL-1α, interleukon-1α; IL-6, interleukon-6; IL-17, interleukon-17; CCL-5, C-C motif chemokine ligand 5; TLRs, toll-like receptor.

**Figure 3 pharmaceuticals-16-00641-f003:**
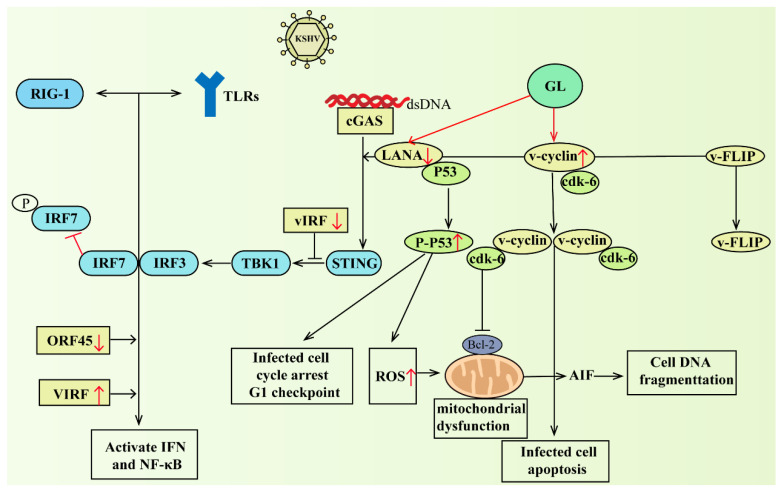
GL downregulated the expression of LANA-1, activated p53, and increased ROS and mitochondrial dysfunction, resulting in G1 cell cycle arrest, DNA fragmentation, and oxidative stress-mediated apoptosis in KSHV-infected cells. On the other hand, it could indirectly attenuate the outer protein ORF45 blocking IRF7 phosphorylation and the activation of type I IFN response, and reduce the inhibitory effect of KSHV-encoded vIRF on the activation of downstream IFN and NF-κB. Abbreviations: RIG-1, retinoicacidinduciblegene-1; TLRs, toll-like receptor; KSHV, Kaposi’s sarcoma-associated herpesvirus; IRF7, interferon regulator factor 7; IRF3, interferon regulator factor 3; TBK1, TANK-binding kinase 1; vIRF, virus interferon regulatory factor; STING, stimulator of interferon genes; cGAS, cyclic GMP-AMP synthase; dsDNA, double-stranded deoxyribonucleic acid; LANA, latency-associated nuclear antigen; FLIP, FLICE inhibitory protein; ORF45, open reading frame 45; ROS: reactive oxygen species; P53, protein 53; cdk-6, cyclin-dependent kinase-6; Bcl-2, B-cell lymphoma-2; AIF, apoptosis inducing factor. The red arrows go up to indicate that GL promotes its expression, and down indicates that GL inhibits its expression.

**Figure 4 pharmaceuticals-16-00641-f004:**
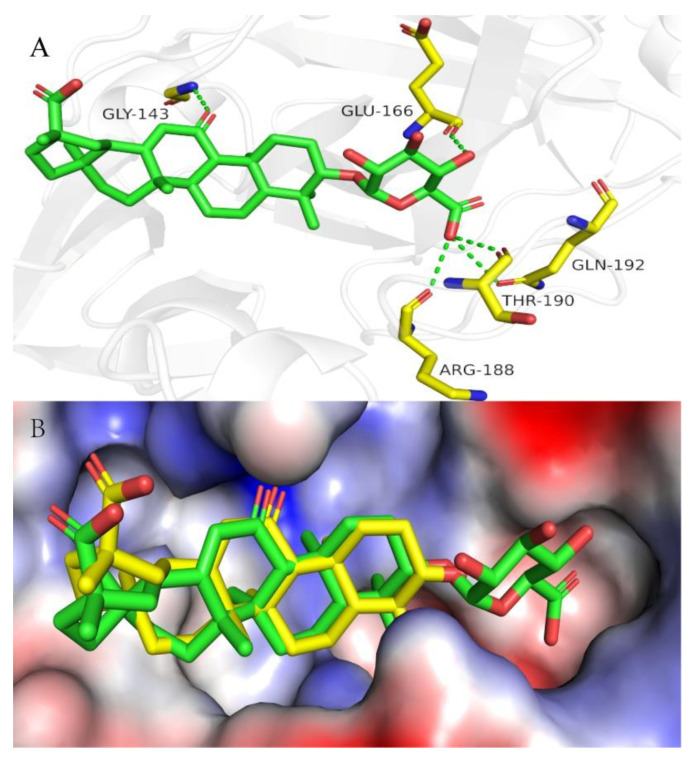
(**A**) Docking model for GAMG with 3CLpro. Close-up view of GAMG (green stick) in the 3CLpro binding pocket (PDB code: 7VU6). Hydrogen bonds are indicated as green dashed lines; only important residues are presented in yellow. (**B**) The superposition diagram of GA and GAMG in the pocket of 3CLpro. Overlay of GA (bright yellow stick) with the GAMG (green stick) in the 3CLpro binding pocket (PDB code: 7VU6). Abbreviations: GLY-143, Glycine-143; GLU-166, Glutamicacid-166; ARG-188, Arginine-188; THR-190, Threonine-190; GLN-192, Glutamine-192.

**Table 1 pharmaceuticals-16-00641-t001:** The effect of GL and its metabolites on the activity of different hepatitis viruses.

Virus	Compound	Involve Mechanisms	Research Subject	Reference
HAV	Glycyrrhizin	Reduce of HAV antigen expression and HAV infectivity.	PLCI PRF/5 cell line	[57]
	Glycyrrhizin	Inhibit an early stage of the HAV replication and HAV penetration of the plasma membrane	PLCI PRF/5 cell line	[58]
HBV	Glycyrrhizin (SNMC)	Suppress the secretion of HBsAg	PLC/PRF/5 cell line guinea pigs	[59,60]
	Glycyrrhizin	Inhibit HBV antigen and anti-inflammatory/activate thymic T cell/immunological regulation	Patients	[61,62,63]
	Glycyrrhizin & entecavir	Decrease serum ALT, AST levels and HBV-DNA concentration	Patient	[64]
	Glycyrrhizin & lamivudine	Suppress HBV-DNA level and normalization of transaminases	Patient (non-Hodgkin’s lymphoma and HBV infection)	[65]
	Glycyrrhetic acid & entecavir	Inhibit MRP4 and BCRP and increased entecavir distribution	HepG2 cell line and rats	[66]
HCV	Glycyrrhizin (SNMC) Glycyrrhizin & PP	Improve serum aminotransferase levels	patients	[67,68,69,70]
	Glycyrrhizin (SNMC)& ursodeoxycholic acid	Improve enzyme abnormalities	patients	[71]
	Glycyrrhizin & IFN (SNMC)	Reduce the incidence of HCC	patients	[72,73]
	Glycyrrhizin (SNMC)	Prevent hepatic steatosis by protecting mitochondria against oxidative stress induced by HCV proteins and iron overload	C57BL/6 transgenic mice	[11]
	Glycyrrhizin & Interferon (IFN)	Inhibit HCV full length viral particles and HCV core gene expression or function	Huh-7 cell line	[74]
	Glycyrrhizin (SNMC)	Inhibit PLA2G1B and affect HCV release	Huh7 and Huh7.5.1 cell lines	[12]
	Glycyrrhizin	prevent membrane penetration of viral particles		[75]
	GA derivatives	Inhibition of pan-HCV genotype entry into human hepatocytes	Huh 7 cell line	[76]
HEV	Glycyrrhizin	Reduce Total bilirubin, ALT and AST	patients	[77]

Abbreviations: HAV, hepatitis A virus; HBV, hepatitis B virus; SNMC, stronger neo-minophagen; HBsAg, hepatitis B surface antigen; ALT, alanine transaminase; AST, aspartate aminotransferase; MRP4, multidrug resistance protein 4; BCRP, breast cancer resistance protein; HCV, hepatitis C virus; IFN, interferon; HCC, hepatocellular carcinoma; PLA2G1B, phospholipase A2 group 1B.

## Data Availability

Not applicable.

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
