# Peer review of "A Review of the Antiviral Activities of Glycyrrhizic Acid, Glycyrrhetinic Acid and Glycyrrhetinic Acid Monoglucuronide"

_pharmaceuticals, 2023, doi:10.3390/ph16050641_

Round 1

Reviewer 1 Report

The topic and review are interesting to the reader, but the author should rearrange them and include well-described, detailed figures or tables. In addition, each section and subsection must be written clearly. Some suggestion:

1)   The title should include "review," such as "A Critical Review of the Pharmacology of Glycyrrhizic Acid, Glycyrrhetinic Acid, and Glycyrrhetinic Acid Monoglucuronide on Antiviral Activity."

2)   The author should include a paragraph or clearly defined section explaining how to get the information, from where, and at what time, as well as the method used to collect the information for the review.

3)   Please cite the figures (1-4) into the text.

4)   Please separate in vitro research from in vivo and clinical studies in the table. Place them at separate tables (Table 1). If utilizing humans or patients, please include the study number, doses, who conducted the research, when, and the outcome.

5)   Please add the subsection (3.3) about Hepatitis E that was already explained in Table 1. Collect additional information that would benefit the reader.

6)   Why does the table not include HSV-1, Variella Zoster Virus, Epstein-Barr Virus, and Kaposi-SahV?

7)   Please study your subsection to determine why it contains the numbers 3.1, 3.2, 4.1, 4.2, 4.3, and 4.4. Where is section 4?

8)   Please compare each virus's morphology, grouping, distribution, etc.

9)   Please describe how each virus works and where glycyrrhizic acid, glycyrrhetinic acid, and glycyrrhetinic acid should be used to target the virus. To compare them, you may use a figure or table.

10)        Figure 4A is oversized. Please adjust

11)        Please read the numerous studies (shown below), including the action mechanism, and cite them.

1)   Elebeedy D, Elkhatib WF, Kandeil A, Ghanem A, Kutkat O, Alnajjar R, Saleh MA, Abd El Maksoud AI, Badawy I, Al-Karmalawy AA. Anti-SARS-CoV-2 activities of tanshinone IIA, carnosic acid, rosmarinic acid, salvianolic acid, baicalein, and glycyrrhetinic acid between computational and in vitro insights. RSC advances. 2021;11(47):29267-86.

2)   Hoever G, Baltina L, Michaelis M, Kondratenko R, Baltina L, Tolstikov GA, Doerr HW, Cinatl J. Antiviral activity of glycyrrhizic acid derivatives against SARS− coronavirus. Journal of medicinal chemistry. 2005 Feb 24;48(4):1256-9.

3)   Frediansyah A, Tiwari R, Sharun K, Dhama K, Harapan H. Antivirals for COVID-19: a critical review. Clinical Epidemiology and global health. 2021 Jan 1;9:90-8.

4)   Langer D, Mlynarczyk DT, Dlugaszewska J, Tykarska E. Potential of glycyrrhizic and glycyrrhetinic acids against influenza type A and B viruses: A perspective to develop new anti-influenza compounds and drug delivery systems. European Journal of Medicinal Chemistry. 2022 Nov 14:114934.

5)   Lin JC. Mechanism of action of glycyrrhizic acid in inhibition of Epstein-Barr virus replication in vitro. Antiviral research. 2003 Jun 1;59(1):41-7.

6)   Frediansyah A, Sofyantoro F, Alhumaid S, Al Mutair A, Albayat H, Altaweil HI, Al-Afghani HM, AlRamadhan AA, AlGhazal MR, Turkistani SA, Abuzaid AA. Microbial Natural Products with Antiviral Activities, Including Anti-SARS-CoV-2: A Review. Molecules. 2022 Jul 5;27(13):4305.

7)   Li JY, Cao HY, Liu P, Cheng GH, Sun MY. Glycyrrhizic acid in the treatment of liver diseases: literature review. BioMed research international. 2014 Oct;2014.

9)   Chen K, Yang R, Shen FQ, Zhu HL. Advances in pharmacological activities and mechanisms of glycyrrhizic acid. Current medicinal chemistry. 2020 Nov 1;27(36):6219-43.

10)        Richard SA. Exploring the pivotal immunomodulatory and anti-inflammatory potentials of glycyrrhizic and glycyrrhetinic acids. Mediators of inflammation. 2021 Jan 7;2021.

11)        Baltina LA, Lai HC, Liu YC, Huang SH, Hour MJ, Baltina LA, Nugumanov TR, Borisevich SS, Khalilov LM, Petrova SF, Khursan SL. Glycyrrhetinic acid derivatives as Zika virus inhibitors: Synthesis and antiviral activity in vitro. Bioorganic & Medicinal Chemistry. 2021 Jul 1;41:116204.

12)        Elebeedy D, Elkhatib WF, Kandeil A, Ghanem A, Kutkat O, Alnajjar R, Saleh MA, Abd El Maksoud AI, Badawy I, Al-Karmalawy AA. Anti-SARS-CoV-2 activities of tanshinone IIA, carnosic acid, rosmarinic acid, salvianolic acid, baicalein, and glycyrrhetinic acid between computational and in vitro insights. RSC advances. 2021;11(47):29267-86.

Author Response

Dear reviewer,

The revised manuscript (changes in the main text were highlighted in yellow color, and only titles were highlighted in yellow color) has been submitted to your journal. For comparing, the reviewers’ suggestions were highlighted in black and our rebuttals are highlighted in red for comparing. We are looking forward to hear your positive response in the near future!

Yours sincerely,

Wenjian Tang

Author Response

(The authors gave the same response as above.)

Reviewer 3 Report

Title:Pharmacology of glycyrrhizic acid, glycyrrhetinic acid and glycyrrhetinic
acid monoglucuronide on antiviral activity”
Authors: Jiawei Zuo, Tao Meng, Yuanyuan Wang, Wenjian Tang

Comments:

The authors describe that due to the lack of effective vaccines and therapeutic drugs, glycyrrhizic acid or glycyrrhizin; glycyrrhetinic acid 3-O-mono-β-D-glucuronide; glycyrrhetinic acid; currently have the therapeutic potential of developmental products as antiviral agents. GL, GA, and GAMG are widely used and generally safe compounds that can be considered primary prevention. While they do not reduce the risk of infection, they may reduce the severity of illness and the burden of medical care.

Some Important points are listed below:

1: The title is very confusing and not attractive because of the so often repeated words. It should be rewritten in principle.

2: The abstract is short and should be rewritten as it contains many repetitions and appears unstructured.

3: The novelty of the article should be much more clearly emphasized, and justification should be given as to why the focus is on the antiviral pharmacological properties and clinical application of GL, GA, and GAMG.

4: The search strategy used for the literature review should be indicated.

5: The limitations associated with use of GL, GA, and GAMG should be more and detailed discussed.

6: All abbreviations should be defined at their first mention and used thereafter.

7: Please use uniform abbreviations throughout the manuscript.

8: English in general needs improvement (often sentences are linguistically incorrect, but often words are simply missing or sentences are incomplete). All text needs to be revised by a native English speaker.

Author Response

(The authors gave the same response as above.)

Reviewer 4 Report

The present article evaluated the implications of the pharmacology of glycyrrhizic acid, glycyrrhetinic acid, and glycyrrhetinic acid monoglucuronide on antiviral activity. The topic is relevant, but there is a need for major changes to improve the initial form:

Shape suggestions

Please revise the instructions for authors and templates because many errors have been identified (i.e., inconsistencies in affiliations and corresponding authors; a list of abbreviations is not necessary because it is abbreviated at the first mention in the abstract and separately treated in the main text, then only the abbreviated form will be used, references in Table I, list of references at the end). Extensive revision is required before being in accordance with all journal rules.

The information in some sections of chapter 2 (the number of lines is not available) is organized in the form of an overly long paragraph, which decreases readability and comprehension. Please reorganize into shorter paragraphs that will be more logical and easier to understand.

There is no need for a blank space between the title of the table and the table per se. The information in the table should not contain punctuation marks. Please revise.

There is no need to have titles in sections or subsections in full capital letters.

The title for section 7 should be reshaped for more specificity.

Content suggestions

The abstract needs to be detailed in order to better understand what unmet needs have been identified and what measures can be taken in terms of biocompound-based pharmacotherapeutic management.

It is advisable to detail in the first two sections aspects related to SARS-CoV-2 infection (mechanism, comorbidities and associated diseases, pharmacological mechanisms) in order to capture current unmet needs and potential combinations between biocompounds and synthetic drugs. I suggest checking and referring to: PMID: 36406478 and PMID: 35131656.

The aim of the paper should be separately presented in the last paragraph of the introduction and needs to be improved from the perspective of describing the contribution to the field under analysis and the elements of scientific novelty presented.

It is advisable to present the conceptual framework for the use of plant-based compounds for the management of microorganism infections. In this respect, it is also necessary to present other species with potential action. I suggest checking and referring to: PMID: 35644118 and PMID: PMID: 35883850.

The algorithm for choosing the proteins involved in the molecular docking study should be explained.

Author Response

(The authors gave the same response as above.)

Round 2

Reviewer 1 Report

I am satisfied with the revised manuscript. However, figures 2 and 3 are still unclear. Please revise/adjust them. Thus, after the revision, the final manuscript will be acceptable to be published in Pharmaceuticals.

Author Response

I am satisfied with the revised manuscript. However, figures 2 and 3 are still unclear. Please revise/adjust them. Thus, after the revision, the final manuscript will be acceptable to be published in Pharmaceuticals.

Response 1): Thank you for your kind suggestions. We have changed Figure 2 and Figure 3 into a clear one in the revised manuscript.

Author Response

Attached file is the point-by-point response to the reviewer’s comments。

Reviewer 4 Report

Check the Instructions for authors regarding 

  • Acronyms/Abbreviations/Initialisms should be defined the first time they appear in each of three sections: the abstract; the main text; the first figure or table. When defined for the first time, the acronym/abbreviation/initialism should be added in parentheses after the written-out form. Proceed consequently for the entire manuscript.
  • Figure 2 is blurred, replace it with a clear one. Moreover, all abbreviations used on the figure must be explained after the title of the figure, below it. Same for Figures 3 and 4. In the case of figure 4, adjust its size (it is huge comparatively with the main text); Also, all the abbreviated amino-acids should be detailed, after the title of the figure.
  • Table 1. Last column. It is enough to keep only the number of the references in brackets, no needed of names and year (they can be easily checked in the References section). The table will look more professional and cleaner.
  • Suggested references have been not properly inserted or not inserted at all. Ref 25 is same with ref. 152. Please check my previous report and proceed consequently.

Author Response

Attached file is the  point-by-point response to the reviewer’s comments.
